# IL-23/IL-17 Axis in Chronic Hepatitis C and Non-Alcoholic Steatohepatitis—New Insight into Immunohepatotoxicity of Different Chronic Liver Diseases

**DOI:** 10.3390/ijms241512483

**Published:** 2023-08-05

**Authors:** Ankica Vujovic, Andjelka M. Isakovic, Sonja Misirlic-Dencic, Jovan Juloski, Milan Mirkovic, Andja Cirkovic, Marina Djelic, Ivana Milošević

**Affiliations:** 1Clinic for Infectious and Tropical Diseases, University Clinical Center of Serbia, 11 000 Belgrade, Serbia; ankica.vujovic88@gmail.com; 2Faculty of Medicine, University of Belgrade, 11 000 Belgrade, Serbia; jovan.juloski@gmail.com; 3Institute of Medical and Clinical Biochemistry, Faculty of Medicine, University of Belgrade, 11 000 Belgrade, Serbia; andjelka.isakovic@med.bg.ac.rs (A.M.I.); sonja.misirlic-dencic@med.bg.ac.rs (S.M.-D.); 4Center of Excellence for Redox Medicine, 11 000 Belgrade, Serbia; 5Zvezdara Medical University Center, Surgery Clinic “Nikola Spasic”, 11 000 Belgrade, Serbia; 6Institute for Orthopedic Surgery “Banjica”, 11 000 Belgrade, Serbia; drckmilan@yahoo.com; 7Department of Medical Statistics, Faculty of Medicine, University of Belgrade, 11 000 Belgrade, Serbia; andja.cirkovic@mfub.bg.ac.rs; 8Institute of Medical Physiology, Faculty of Medicine, University of Belgrade, 11 000 Belgrade, Serbia

**Keywords:** chronic hepatitis C, non-alcoholic steatohepatitis, IL-23/IL-17 axis, immunohepatotoxicity

## Abstract

Considering the relevance of the research of pathogenesis of different liver diseases, we investigated the possible activity of the IL-23/IL-17 axis on the immunohepatotoxicity of two etiologically different chronic liver diseases. A total of 36 chronic hepatitis C (CHC) patients, 16 with (CHC-SF) and 20 without significant fibrosis (CHC-NSF), 19 patients with non-alcoholic steatohepatitis (NASH), and 20 healthy controls (CG) were recruited. Anthropometric, biochemical, and immunological cytokines (IL-6, IL-10, IL-17 and IL-23) tests were performed in accordance with standard procedure. Our analysis revealed that a higher concentration of plasma IL-23 was associated with NASH (*p* = 0.005), and a higher concentration of plasma IL-17A but a lower concentration of plasma IL-10 was associated with CHC in comparison with CG. A lower concentration of plasma IL-10 was specific for CHC-NSF, while a higher concentration of plasma IL-17A was specific for CHC-SF in comparison with CG. CHC-NSF and CHC-SF groups were distinguished from NASH according to a lower concentration of plasma IL-17A. Liver tissue levels of IL-17A and IL-23 in CHC-NSF were significantly lower in comparison with NASH, regardless of the same stage of the liver fibrosis, whereas only IL-17A tissue levels showed a difference between the CHC-NSF and CHC-SF groups, namely, a lower concentration in CHC-NSF in comparison with CHC-SF. In CHC-SF and NASH liver tissue, IL17-A and IL-23 were significantly higher in comparison with plasma. Diagnostic accuracy analysis showed significance only in the concentration of plasma cytokines. Plasma IL-6, IL-17A and IL-23 could be possible markers that could differentiate CHC patients from controls. Plasma IL-23 could be considered a possible biomarker of CHC-NSF patients in comparison with controls, while plasma IL-6 and IL-17-A could be biomarkers of CHC-SF patients in comparison with controls. The most sophisticated difference was between the CHC-SF and CHC-NSF groups in the plasma levels of IL-10, which could make this cytokine a useful biomarker of liver fibrosis.

## 1. Introduction

Nowadays, the most common chronic liver diseases (CLD) have been attributed to the non-alcoholic fatty liver disease (NAFLD) (59%), followed by the hepatitis B virus (HBV) (29%), hepatitis C virus (HCV) (9%) and alcoholic liver disease (ALD) (2%). Less than 1% of cases are caused by other liver conditions, such as primary biliary cholangitis, primary sclerosing cholangitis, alpha-1-antitrypsin deficiency, Wilson’s disease and autoimmune hepatitis. Historically, viral hepatitis has been the leading etiology for CLD, but improved prevention strategies in the case of hepatitis B and treatment in the case of hepatitis C have led to changing CLD trends [1]. The absolute number of CLD cases (inclusive of any stage of disease severity) worldwide is estimated at 1.5 billion [2]. CLD caused 1.32 million deaths in 2017, approximately two-thirds among men and one-third among women [1]. In 2019, CLD and cirrhosis were the ninth leading cause of death for males (1.9%) but were not even in the top 10 for females. For those aged 25–44, CLD and cirrhosis were among the leading causes of mortality [3].

The role of the immune system in CLD was recognized decades ago [4,5], but not all immunological mechanisms of liver damage have been clarified to date, which could be potential therapeutic targets. Interleukin-23 (IL-23) and its downstream factor IL-17 are described as the key cytokines involved in immunopathogenesis of CLD [6]. IL-23/IL-17 axis’s role is well known in other chronic diseases, such as rheumatic diseases, inflammatory bowel diseases, chronic inflammatory skin diseases and chronic lymphocytic leukemia [7,8,9,10]. In chronic inflammation, antigen-stimulated dendritic cells and macrophages produce IL-23, which promotes the development of Th17/ThIL-17 cells. Th17/ThIL-17 cells produce IL-17, which enhances T-cell priming and triggers potent inflammatory responses by inducing the production of a variety of inflammatory mediators. IL-23 also acts on dendritic cells and macrophages in an autocrine/paracrine manner to stimulate the generation of proinflammatory cytokines, such as IL-1, IL-6, IL-17 and TNF-α. IL-12–stimulated Th1 cells produce IFN-γ and suppress the differentiation of Th17/ThIL-17 cells. Th1 cells may play an immunoregulatory, protective role in the development of chronic inflammation [11,12]. Numerous studies described the influence of the IL-23/17 axis on the development of liver fibrosis with heterogeneous results [6,13]. These results mostly described pathophysiology but also addressed potential immunotherapeutic options [14,15]. The most contradictory results are related to chronic hepatitis C (CHC) and non-alcoholic steatohepatitis (NASH).

The introduction of direct-acting antivirals (DAAs) has changed the paradigm of HCV infection and enabled CHC to become curable. Although DAAs have significantly improved the rate of sustained virological response, disease progression may not be completely avoidable, especially in patients with a high degree of fibrosis [16]. This is the reason for the continuous need for research on various factors, especially immunological ones, that characterize CHC. The immune response to HCV infection has a specific role in the enhancement of hepatic fibrogenesis. Multiple growth factors, inflammatory cytokines and chemokines may regulate the activation of hepatic stellate cells (HSCs) and their transformation to myofibroblasts. [17]. Although the pathophysiology of HCV is not entirely understood, interleukin (IL)-17-producing CD4+ T cells, also named T-helper 17 cells (Th17), have been shown to play a significant role [18]. In addition, Rios et al.’s research [19] revealed that the only lymphocyte subset linked to advanced fibrosis was Th17. However, it is still unknown if Th17 can play both a positive and a negative role. Furthermore, it is still unclear whether there are variations in the concentration of proinflammatory cytokines at local and systemic levels in CHC that can potentially direct immunotherapeutic strategies in these patients.

In NASH, inflammation is one of the key drivers of disease progression and fibrosis development. Regulation of inflammation is controversial, depending on multiple intrahepatic and extrahepatic factors. According to research by Tang et al., NASH patients had considerably higher levels of the hepatic expression of Th17 cell-related genes (retinoid-related orphan receptor gamma (ROR)γt, IL-17, IL-21 and IL-23) than healthy controls. Th17 cells and IL-17 were associated with hepatic steatosis and proinflammatory response in NASH and facilitated the transition from simple steatosis to steatohepatitis [20]. Heredia et al. demonstrated that IL-23 receptor deletion does not impact liver inflammation and fibrosis in the choline-deficient, L-amino acid-defined and high-fat diet-induced NASH model. These results led to the hypothesis that IL-23 signaling is not essential for NASH pathogenesis in preclinical models and that blocking this pathway may not be a successful therapeutic strategy to slow the disease progression in NASH patients [13]. Therefore, cellular and molecular targets have been discovered in NASH through research in recent years, but it has been difficult to translate these discoveries into disease-modifying therapies.

In order to elucidate the pathogenesis and explore potential diagnostic and immunotherapeutic pathways, this study was conducted to investigate if there are differences in the effect of the IL-23/IL-17 axis in two chronic liver diseases with different etiopathogenesis—CHC and NASH.

## 2. Results

All demographic, anthropometric and biochemical characteristics of study subjects are shown in Table 1. The values of metabolic syndrome (MS) parameters such as body mass index (BMI), body mass (BM), cholesterol, low-density lipoprotein cholesterol (LDL) and triglycerides were significantly higher in NASH in comparison with all the other groups, while high-density lipoprotein cholesterol (HDL) levels were significantly lower in all of the liver diseases groups in comparison to control group (CG). In the chronic hepatitis C group with significant fibrosis (CHC-SF), albumin and platelet (PLT) levels were significantly lower in comparison with the chronic hepatitis C group with non-significant fibrosis (CHC-NSF), NASH and CG. Also, in the CHC-SF group, alpha fetoprotein (AFP) was higher in comparison with the other three groups. Aspartate aminotransferase (AST) and alanine aminotransferase (ALT) levels were significantly higher in all liver diseases groups in comparison with CG, as well as in the CHC-SF group in comparison with the CHC-NSF and NASH groups.

The levels of IL-6 in plasma were not significantly different between the compared groups (*p* = 0.075), but a difference was noted in liver tissue (*p* = 0.043). CG had the lowest measured plasma levels (median 3.91 pg/mL), with 8 out of 20 samples having levels below the level of detection (LOD). Median levels of plasma IL-6 were 12.60, 7.30 and 10.11 pg/mL, and median liver tissue IL-6 levels were 49.35, 23.06 and 43.63 pg/mL for the NASH, CHC-NSF and CHC-SF groups, respectively. Post-hoc analysis showed that CHC-SF and NASH had significantly higher levels of plasma IL-6 in comparison with CG. The liver tissue levels were significantly higher in comparison with plasma IL-6 levels in the CHC-SF and NASH groups (Figure 1).

The median concentration of IL-10 in plasma showed a clear decreasing trend from healthy controls to NASH, as well as from hepatitis without significant fibrosis to liver cirrhosis. The differences between the groups were statistically significant (*p* < 0.0001), while the post-test revealed the difference between CG and both the CHC-NSF and CHC-SF groups, as well as between NASH and both the CHC-NSF and CHC-SF groups. Also, levels in CHC-SF were statistically lower in comparison with the CHC-NSF group. The levels of IL-10 in liver tissue were not significantly different between the compared groups (*p* = 0.162), but the post-test showed a significantly higher median value in CHC-NSF in comparison with NASH (*p* < 0.05). The median plasma levels were 162.6, 137.5, 79.44 and 59.39 pg/mL for the CG, NASH, CHC-NSF and CHC-SF groups, respectively. Also, the median tissue levels were 61.96, 43.26, and 46.85 pg/mL for the NASH, CHC-NSF and CHC-SF groups, respectively. The liver tissue levels were significantly lower in comparison with plasma IL-10 levels in the CHC-NSF and NASH groups (Figure 2).

The plasma concentrations of IL-17A were different between the groups (*p* = 0.000). The median plasma level in the CHC-SF group was 17.79 pg/mL, while the median concentration in the NASH group was 26.56 pg/mL. The majority of patients from the CHC-NSF as well as CG groups had levels below the LOD, and thus, median serum levels were considered as 0. The plasma levels of IL-17A in the CHC-SF and CHC-NSF groups were lower in comparison with the group of patients with NASH (*p* = 0.0035, *p* = 0.000, respectively). In comparison with CG and CHC-NSF, the plasma levels of IL-17A were higher in CHC-SF and NASH (*p* = 0.000, *p* = 0.000, *p* = 0.000, respectively) (Figure 3). Median tissue levels were 49.35, 23.06, 43.63 and 0 pg/mL for NASH, CHC-NSF, CHC-SF and CG groups, respectively. The differences between the groups were statistically significant (*p* = 0.017), while the post-test revealed the differences between CHC-NSF and CHC-SF, as well as the CHC-NSF and NASH groups (*p*= 0.050, for all). The liver tissue levels were significantly higher in comparison with plasma IL-17A levels in the CHC-SF and NASH groups (Figure 3).

Plasma concentrations of IL-23 were different between the study groups (*p* = 0.0105) (Figure 4). The control samples had the lowest median value (66.15 pg/mL), and the concentration of plasma IL-23 was gradually higher in CHC-SF and CHC-NSF (median values were 88.72 pg/mL and 92.64 pg/mL, respectively), and finally, patients with NASH had the highest plasma concentrations of IL-23 (131.80 pg/mL). In addition, the difference between the levels in CG and all of the other groups (CHC-NSF, CHC-SF and NASH), as well as between NASH and CHC-SF, were significant (*p* < 0.05). The levels of IL-23 in the liver tissue were not significantly different between the compared groups (*p* = 0.093), but there was a post-test significant difference in liver tissue between CHC-NSF and NASH (*p* = 0.042). The liver tissue levels of IL-23 were significantly higher in comparison with plasma in the CHC-SF and NASH groups. The median liver tissue for NASH group was 49.35 pg/mL, for CHC-NSF 23.06 pg/mL, and for CHC-SF 43.63 pg/mL.

The analysis of the association between cytokine levels and both laboratory and biochemical parameters within the study groups was performed. When we analyzed all patients with CHC, serum levels of AST and ALT displayed a strong positive correlation with serum IL-17A levels (ρ = 0.716, *p* < 0.01; ρ = 0.691, *p* < 0.01, respectively), as well as a moderate positive correlation with tissue IL-17A levels (ρ = 0.393, *p* < 0.05; ρ = 0.377 *p* < 0.05, respectively). Also, in CHC patients, serum IL-17A levels have shown negative moderate correlation with albumins (ρ = −0.646, *p* < 0.01) and fibrinogen (ρ = −0.363, *p* < 0.01). Tissue levels of IL-17A and IL-23A display a moderate positive correlation with AP (ρ = 0.399, *p* < 0.05). In the CHC-NSF subgroup, we obtained a moderate positive association between CRP and tissue IL-10 level (ρ = 0.639, *p* = 0.003) and between total cholesterol and tissue IL-23 level (ρ = 0.463, *p* = 0.046), while there was moderate negative association between fibrinogen and plasma IL-6 levels (ρ = −0.444, *p* = 0.050). In the CHC-SF study subgroup, plasma IL-6 level was in a moderate positive association with TSH (ρ = 0.577, *p* = 0.019), and plasma IL-10 level was in a moderate positive association with hemoglobin and TSH (ρ = 0.577, *p* = 0.019 and ρ = 0.533, *p* = 0.034, respectively). In the same study subgroup, we found a moderate negative association between plasma IL-17A and AP (ρ = −0.649, *p* = 0.006) and a positive moderate association between plasma IL-23 and the number of platelets (ρ = 0.559, *p* = 0.024). The levels of AP were in moderate positive association with tissue levels of IL-17A and IL-23 (ρ = 0.687, *p* = 0.007 and ρ = 0.676, *p* = 0.008, respectively). The NASH subgroup was specific according to a moderate positive association between plasma IL-10 levels and CRP and urea (ρ = 0.509, *p* = 0.026 and ρ = 0.567, *p* = 0.014), plasma IL-17A levels and LDL (ρ = 0.458, *p* = 0.049), and plasma IL-23 levels and urea (ρ = 0.556, *p* = 0.017). The tissue levels of IL-6 were in moderate negative association with direct bilirubin and albumins (ρ = −0.638, *p* = 0.014 and ρ = −0.642, *p* = 0.013, respectively). Tissue IL-17A was in moderate negative association with albumins, too (ρ = −0.614, *p* = 0.015). Tissue IL-23 was in moderate negative association with ALT (ρ = −0.583, *p* = 0.023).

In order to distinguish the main study groups (CHC and NASH) from the healthy population (CG), as well as to determine the difference between the CHC subgroups (CHC-NSF and CHC-SF) according to plasma/tissue cytokine concentrations, logistic regression analysis was performed, and the results are presented in Table 2. Higher concentrations of plasma IL-23 were associated with NASH (*p* = 0.005), and higher concentrations of plasma IL-17A and lower concentrations of plasma IL-10 were associated with CHC (*p* = 0.001 and *p* = 0.016, respectively) in comparison with healthy controls. Also, lower concentrations of plasma IL-10 were specific for CHC-NSF, while higher concentrations of plasma IL-17A were specific for CHC-SF in comparison with healthy controls (*p* = 0.012 and *p* = 0.005, respectively). CHC patients, as well as CHC-NSF and CHC-SF patients, were distinguished from NASH patients according to lower concentrations of plasma IL-17A (*p* = 0.007, *p* = 0.004 and *p* = 0.021, respectively). Finally, a useful parameter for the differentiation of CHC-NSF and CHC-SF patients was IL-10 in plasma. Lower concentrations of plasma IL-10 were associated with CHC-SF (*p* = 0.013). All conclusions remained significant after adjustment for age and gender.

Finally, in order to find possible biomarkers among plasma and tissue cytokine levels for differentiation between evaluated subgroups, diagnostic accuracy analysis was assessed (Table 3). Cytokines were considered as possible biomarkers if the area under the curve was at least 50% and with the highest sum of sensitivity (Sn) and specificity (Sp) according to Youden’s rule. Plasma IL-6, IL-17A and IL-23 were possible markers that could differentiate CHC patients from controls. Plasma IL-23 could be regarded as possible biomarker of CHC-NSF patients in comparison with controls, while plasma IL-6 and IL-17A could be biomarkers of CHC-SF patients in comparison with controls. CHC-NSF and CHC-SF patients had the most sophisticated difference, and it might be found according to the level of plasma IL-10 cytokine as a biomarker of fibrosis. Therefore, the ROC curve is presented for this parameter (Figure 5). There were no significant potential markers for differentiation between other subgroups.

## 3. Discussion

Considering the relevance of pathogenesis research of different liver diseases, this study investigated the possible activity of the IL-23/IL-17 axis on the immunohepatotoxicity of two etiologically different CLDs. NAFLD, recently renamed as metabolic (dysfunction)-associated fatty liver disease (MAFLD), is a complex, multifactorial disease that includes one or more components of the MS, like systemic hypertension, dyslipidemia, visceral obesity, insulin resistance or diabetes [21,22]. Disease progression from non-alcoholic fatty liver (NAFL) is characterized by macrovesicular hepatic steatosis that may be accompanied by mild inflammation due to NASH, which is additionally characterized by the presence of inflammation and cellular injury (ballooning), with or without fibrosis. Liver cirrhosis represents the end stage of chronic liver disease, which is characterized by bands of fibrous septa leading to the formation of cirrhotic nodules. Further progression leads to irreversible damage, decompensation of liver function, and hepatocellular carcinoma. The prevalence of NASH has risen rapidly worldwide. In the last three decades, these rates have increased from 25% to 38%, and NASH is likely to become the main indication for liver transplantation in the near future [23,24,25,26]. On the other hand, CHC is an infectious disease, caused by HCV infection. The acute HCV infection rarely resolves spontaneously. Approximately 80% of infected individuals become chronic carriers, of which 10–20% of patients will develop liver cirrhosis and 1–5% will develop hepatocellular carcinoma (HCC) within 20–30 years. [27].

Both of these liver diseases, CHC and NASH, as one of the first laboratory abnormalities, show elevated transaminase levels as a result of liver damage. NASH determines asymptomatic elevation of ALT and AST levels in up to 90% of cases, once other liver disease causes are excluded [28]. Numerous studies have shown that serum levels of ALT and AST are elevated in CHC patients in different stages of the disease [29,30]. In this study, AST and ALT levels were significantly higher in all liver disease groups in comparison with CG. Also, higher transaminase values were in the CHC-SF group in comparison with the CHC-NSF and NASH groups, which is expected because of the higher levels of liver fibrosis and necrosis of hepatocytes in this group (F3/F4 in CHC-SF in comparison with F0/F1/F2 in the CHC-NSF and NASH groups). Some values of MS parameters such as BM, BMI, cholesterol, LDL and triglycerides were significantly higher in NASH compared to all the other groups, while HDL levels were significantly lower in all of the liver disease groups compared to CG. These results are expected and consistent with previous studies [31,32]. Patients from the CHC-SF group have shown lower levels of albumins and PLT in comparison with CHC-NSF, NASH and CG, which is an indicator of the impaired synthetic liver function in the advanced stage of the disease. Also, this group of patients has shown significantly higher levels of AFP in the absence of HCC, compared to other liver disease groups, which is all in accordance with the published data [33,34,35,36]. Other biochemical parameters did not show any statistical significance.

From the immunopathogenic aspects, the literature data show controversial results about both of these chronic liver diseases. In liver diseases, before cirrhosis is established, multiple pathways of fibrogenesis are activated as a result of continuous interaction between pathogen-related factors, the host genetics, such as certain HLA haplotypes and cytokines gene polymorphisms, liver resident cells and the immune system [37,38,39,40]. We focused on the IL-23/IL-17 axis, one of the important proinflammatory responses to chronic stimulation. In CHC immunopathogenesis, there are two possible pathways. The first mechanism is the production of the thymic stromal lymphopoietin (TSLP) by HCV-infected hepatocytes, in an NF-κB-dependent process, leading to the activation of antigen-presenting cells (APCs) which produce IL-1, IL-6, IL-21 and IL-23. The second mechanism includes the toll-like receptor 2 ligand, promoting, by itself, the activation of the APCs to produce inflammatory cytokines that favor Th17 differentiation. After being differentiated, the Th17 cells secrete their cytokines (IL-17, IL-21 and IL-22), with IL-17 being the main driver of a chain of events that stimulate proinflammatory and profibrotic pathways. IL-17 induces the transformation of HSCs to myofibroblasts and the epithelial–mesenchymal transition of the hepatocytes, promoting the synthesis of the extracellular matrix, cell contractility, and all changes in the liver microstructure and microcirculation which lead to progression of fibrosis [41]. The role of the IL-23/IL-17 axis in the pathogenesis of liver disease has been extensively evaluated in multiple mouse models of liver injury. Yan et al. have shown that genetic deletion of IL-17RA or IL-17A or antibody-mediated neutralization of IL-17A can provide protection from induced hepatitis and also that levels of IL-17 and IL-6 increase in parallel with the severity of liver injury reflected by ALT and histological assay [14]. Meng et al. have demonstrated that the IL-23/Th17 axis plays an important role in the antiviral response in chronic HCV infection. IL-23 may enhance the antiviral activity of interferon-based therapy by modulating the expression of Th17 cell-associated molecules in HCV-infected patients [6]. In our study, CHC-SF patients as well as NASH patients had significantly higher plasma levels of IL-17A in comparison with CHC-NSF and CG, while all three CLD groups had higher levels of IL-23 in comparison with CG. In addition, IL-17A plasma levels in CHC-NSF and CG were below the LOD, so in these two groups we used 0 pg/mL for median values. This approach has been previously used in a study which investigated systemic sclerosis, where CG and lower severity diseases group had levels of IL-17A below the LOD and thus have used 0 pg/mL as the median in order to perform statistical analysis [42]. Previous studies have shown that patients with CHC have increased levels of IL-17A and IL-23 in comparison with healthy group, but these studies have not included the degree of liver fibrosis in the analysis [6]. Also, recent studies have shown that levels of these cytokines are increased in NASH patients in comparison with healthy group [20,43] which is all in correlation with our results. We also detected significantly higher plasma levels of IL-17A and IL-23 in the NASH group in comparison with the CHC-SF group, regardless of the higher level of fibrosis in the CHC-SF group. This result may lead to the conclusion that the IL-23/IL-17 axis is activated at a higher systemic level in NASH compared to CHC patients. Plasma levels of IL-6 were significantly higher in the CHC-SF and NASH groups in comparison with CG. According to the research by Souza-Cruz et al., CHC patients with severe liver fibrosis (F3/F4) had considerably higher serum levels of proinflammatory cytokines such as IL-6 and IL-17 [44]. In addition, Rau et al. have shown that in NASH patients, levels of IL-6 are increased in comparison with a healthy group [45]. Both of these results are in accordance with the results obtained in our study. We did not find differences in IL-6 and IL-23 plasma levels between the two groups with CHC, but levels of IL-17A were significantly higher in the CHC-SF group than in CHC-NSF group. The importance of the increased levels of IL-17 in the different stages of CHC was previously recognized [46,47] and seems to play an important role even in the progression of CHC to HCC [48].

IL-10, the only anti-inflammatory cytokine in our study, was significantly higher in CG in comparison with both of the CHC groups, whereas the level was higher in CHC-NSF in comparison with the CHC-SF group. One of the previous studies has shown decreased levels of IL-10 in the healthy group compared to the chronic hepatitis group [49], which is not in correlation with our results. On the other hand, Dorcas et al. demonstrated that twenty-five percent (41/163) of the anti-HCV positive participants had recovered from HCV, and these patients had significantly higher concentrations of IL-10 compared to those with an active HCV infection [50]. Our results can be explained by an increased anti-inflammatory response in CHC without significant fibrosis because it is obvious that in advanced fibrosis and cirrhosis the dominant component of the immune response becomes proinflammation. In the NASH group, levels of IL-10 were higher in comparison with both groups of CHC patients, but there was no difference between NASH and CG in our study. One of the studies on rats has shown higher serum levels of IL-10 in the NASH-induced group compared to the control group [51], but Vonghia el al. [31] had not shown a statistical difference in serum IL-10 levels in NASH and CG, which is in correlation with our study.

Liver tissue levels of all of these cytokines were measured only in CLD groups. Significantly higher tissue levels of IL-6 and IL-17A were found in CHC-SF compared to the CHC-NSF group. Aboushousha et al. demonstrated a positive correlation between IL-17 liver tissue expression with marked grades of hepatitis activity and high scores of liver fibrosis compared to lower ones [52]. Also, Tan et al. reported that intrahepatic IL-17 expression was positively correlated with serum indices of hepatic fibrosis [53]. The study by Gomes et al. suggested that IL-17 in CHC may initiate steatohepatitis progressing to HCC [54]. All of these results are in correlation with the results of our study and demonstrate a strong association of IL-17A with chronic hepatitis-induced liver fibrosis, but more preclinical and patients cohort studies are needed to confirm this link. We have found significantly lower tissue levels of IL-6, IL-17A, IL-23 and IL-10 in CHC-NSF in comparison with the NASH group, despite the same level of fibrosis in these two groups (F0/F1/F2). To the best of our knowledge, there is no available data comparing tissue levels of these cytokines in NASH and CHC patients, but a possible explanation is that the local liver immunopathogenetic response in NASH patients leads to stronger local inflammation and probably faster liver damage with consequent cirrhosis, compared with CHC patients.

Differences between plasma and liver tissue cytokine levels in each of the CLD groups were also analyzed. In CHC-NSF, there were no significant differences between plasma and liver tissue levels of IL-6, IL-17A and IL-23, whereas the tissue level of IL-10 was lower compared to the plasma level. On the other hand, in the CHC-SF group as well as in NASH group, there were increased liver tissue levels of IL-6, IL-17A and IL-23 in comparison with plasma levels, which could represent a higher local than systemic immunology response. Previous studies have shown that the liver IL-6 expression is correlated positively with the severity of inflammation and the degree of fibrosis observed in NASH patients, coupled with the fact that saturated FFA has been demonstrated to induce IL-6 production by hepatocytes in vitro [55]. Beihui et al. as well as Li et al. have demonstrated that the levels of Th17 cell-related cytokines [IL-6, IL-17 and IL-23) in serum and in liver tissue were increased in experimental animal models of NASH-induced disease compared to a healthy group, but they have not analyzed differences between serum and liver tissue levels [56,57]. A recent study has described increased proportions of both circulating and liver-infiltrating Th17 cells in CHC patients compared to healthy individuals, and both measures of Th17 cells were correlated with the severity of liver inflammation [58]. The study of Macek Jilkova et al. concluded that in chronic hepatitis, the number of IL-17(+) neutrophils in fibrotic septa and portal areas is strongly correlated with the stages of fibrosis, contributing significantly to the total IL-17 production in liver tissue [59]. All of these results, as well as the results from our study, indicate strong local hepatic inflammation in patients with CHC with significant fibrosis and NASH patients.

Logistic regression analysis has confirmed that higher plasma concentrations of IL-23 were associated with NASH, whereas higher IL-17A and lower IL-10 plasma concentrations were associated with CHC in comparison with healthy controls. Lower concentrations of plasma IL-10 were specific for CHC-NSF, while higher concentrations of plasma IL-17A were specific for CHC-SF in comparison with healthy controls. Finally, diagnostic accuracy analysis assessed plasma IL-6, IL-17A and IL-23 as possible markers that could differentiate CHC patients from controls. Plasma IL-23 could be regarded as possible biomarker of CHC-NSF patients in comparison with controls, while plasma IL-6 and IL-17A could be biomarkers of CHC-SF patients in comparison with controls. The most sophisticated difference was in higher levels of IL-10 in CHC-NSF in comparison with CHC-SF. This analysis assessed IL-10 as a biomarker of fibrosis in CHC, which is also suggested by previous studies [49,60].

In the CHC groups, serum levels of AST and ALT display a strong positive correlation with serum IL-17A levels, as well as a moderate positive correlation with tissue IL-17A levels. Also, in the CHC groups, serum IL-17A levels have shown negative moderate correlation with albumins and fibrinogen, but on the other hand, tissue levels of IL-17A and IL-23 display a moderate positive correlation with AP. Elbanan et al. [61] described a positive correlation between serum IL-17 levels and transaminases as well as negative correlation between IL-17 and serum albumins, which is in correlation with our results. Serum levels of IL-17A have shown a positive moderate correlation with LDL in the NASH group. One of the previous studies investigated whether LDL can directly affect hepatic inflammation in mice. Mice were injected with a bolus of oxLDL and sacrificed after 2, 6 and 24 h of injection. After oxLDL injection, they found that lysosomal trapping of oxLDL was correlated with elevated expression of hepatic inflammatory genes [62]. So, these data suggest a causal relationship between LDL and hepatic inflammation and provide new bases for prevention and treatment of NASH.

## 4. Materials and Methods

### 4.1. Subjects

A total of 75 subjects were included in this study. Among these, 36 patients with CHC infection and 19 patients with NASH were recruited from the Clinic for Infectious and Tropical Diseases, University Clinical Center of Serbia (UCCS) from October 2018 to December 2019. The diagnosis for the CHC patients was made using the European Association for the Study of the Liver (EASL) Clinical Practice Guidelines: Management of hepatitis C virus infection [63] and included anti-HCV antibodies during at least 6 months with positive real-time polymerase chain reaction (PCR) assays for quantifying HCV RNA. After histological verification from liver tissue obtained by liver biopsy, patients with CHC were divided into two groups—the group with non-significant fibrosis- CHC-NSF (F0/F1/F2) with 20 patients, and the group with significant fibrosis/cirrhosis- CHC-SF (F3/F4) with 16 patients.

The diagnosis of NASH was made using the EASL Clinical Practice Guidelines for the management of non-alcoholic fatty liver disease [64]. After histological verification from liver tissue obtained by liver biopsy, in the NASH group were included 19 patients. In addition, 20 age- and sex-matched healthy donors were used as controls. Peripheral blood samples were collected from all healthy controls and patients with CHC and NASH, while fine needle liver biopsy was done in patients with CHC and NASH. Participants with the following conditions were excluded:Pregnancy;Presence of decompensated cirrhosis;Co-infection with human immunodeficiency virus (HIV) and co-infection with hepatitis A, B, or D virus for CHC group of patients; infection with HIV, hepatitis A, B, C or D for NASH and control group of patients;Other chronic or acute liver disease (autoimmune/toxic);Presence of any of an immunocompromised state;Patients with HCC.

Written informed consent was obtained from all of the patients. The study protocol was approved by the ethics committees of the Clinic for Infectious and Tropical Diseases, University Clinical Center of Serbia. Our research was subject to ethical standards that promote and ensure respect for all human subjects and protect their health and rights by the Declaration of Helsinki.

### 4.2. Pathohistological Analysis of Liver Tissue

Using Menghini needles, liver biopsies were done in the Clinic for Infectious and Tropical Diseases and after that were processed by the Institute of Pathology at the University of Belgrade. Liver biopsy specimens were fixed in formalin to form cross-links and prevent degradation. The METAVIR score was used to classify the liver fibrosis in CHC patients: F0—No fibrosis; F1—Portal fibrosis without septa; F2—Portal fibrosis with few septa; F3—Numerous septa without cirrhosis; F4—Cirrhosis [65]. In NAFLD patients, we used the American Association for the Study of Liver Diseases (AASLD) scoring system [66]:Steatosis was graded as follows: <5% of liver parenchyma—0; 5–33%—1; >33–66%—2; >66%—3;Fibrosis was staged: none—0; perisinusoidal or periportal—1; perisinusoidal and portal/periportal—2; bridging fibrosis—3; cirrhosis—4;Inflammation: lobular, portal (0–3);Hepatocellular ballooning degeneration (0–2).

### 4.3. Detection of HCV Antibody, Viral Load and Genotypes of HCV

Virological analyses were done in the Virology Laboratory of the Clinic for Infectious and Tropical diseases. Anti-HCV antibodies were detected by ELISA test (Roche, Branchburg, NJ, USA), and an HCV genomic RNA test (HCV-RNA) was performed in the Amplicor RT-PCR (Roche, NJ, USA) that presents a sensitivity of 50 IU/mL. Samples with detectable HCV-RNA were further genotyped through in-house RT-nested PCR and RFLP analysis, and the viral load was determined by RT-PCR (Amplicor HCV Monitor, Roche, NJ, USA) with data expressed as IU/mL.

### 4.4. Biochemical Assays

Biochemical analyses (Erythrocytes (cells/mL), Hemoglobin (g/L), Leucocytes (cells/mL), Thrombocytes (cells/mL), Glucose (mmol/L), Uric acide (µmol/L), Creatinine (µmol/L), Total cholesterol (mmol/L), Triglycerides (mmol/L), Total proteins (g/L), Albumin (g/L), Total bilirubin (mg/dL), Direct bilirubin (mg/dl), Fibrinogen (mg/dL), Aspartate aminotransferase (U/L), Alanine aminotransferase (U/L), γ-glutamyl transpeptidase (U/L); Alkaline phosphatase (U/L); Alpha fetoprotein (ng/mL); Thyroid-stimulating hormone (microU/mL); Low-density lipoprotein cholesterol (mmol/L); High-density lipoprotein cholesterol (mmol/L); INR (international normalized ratio); and C-reactive protein (mg/dL)) were observed in the Biochemical Laboratory of the Clinic for Infectious and Tropical Diseases using standard methods.

### 4.5. Measurement of Plasma and Tissue Cytokines (IL-6, IL-10, IL-17A and IL-23)

Previously collected plasma from the CG, NASH, CHC-NSF and CHC-SF groups was used for the measurement of IL-6, IL-10, IL-17 and IL-23 using Human ELISA kits from Elabscience (Houston, TX, USA) (#E-EL-H0102, #E-EL-H0103, # E-EL-H0105 and # E-EL-H0107, respectively). The same cytokines were measured in liver biopsy tissues using the same ELISA kits in NASH, CHC-NSF and CHC-SF. Upon liver biopsy, the tissues were weighted and snap frozen. Each sample was mechanically homogenized in lysis buffer for the extraction of proteins (150 mM NaCl, 25 mM Tris-HCl pH 7.6, 1% IGEPAL, 0.1% SDS, 1% sodium deoxycholate, 1 mM EDTA) through repeated freeze-thaw cycles. The samples were centrifuged at 14,000× *g* at 4 °C for 15 min and supernatants were used for ELISA. The assays were performed according to manufacturers’ instructions (Elabscience). In short, ELISA plates are pre-coated with specific antibody, a biotinylated antibody for a specific cytokine is used for detection, and avidin-horseradish peroxidase conjugate is used to convert the substrate color into blue. The enzyme-substrate reaction is terminated by the addition of a stop solution. The intensity of the developed blue color is directly proportional to the concentration of the measured cytokine. The absorbance (optical density, O.D.) was measured at a wavelength of 450 nm on an automated plate reader (Tecan, Dorset, UK). The concentrations of the cytokines were calculated by comparing the O.D. values of samples with standards using a four-parameter logistic curve in GraphPad Prism. The results for sera are expressed as pg/mL, while the results for tissues are expressed as pg/mL/mg of wet tissue weight. All of these measurements were done at the Institute of Medical and Clinical Biochemistry, Faculty of Medicine, University of Belgrade.

### 4.6. Anthropometric Measurements

BM was measured under a standard procedure on Tanita body analyzer (In body 970, Seoul, South Korea). A SECA–217 stadimeter was used for BH (Hamburg, Germany). BMI was calculated as BM in kg/(BH in m)^2^.

### 4.7. Statistical Analysis

The data obtained are presented as mean (± S.E.M.) or median (interquartile range). The Shapiro–Wilk test was used for an assessment of the normality of continuous data. The differences between the groups were assessed using an ANOVA parametric test for multiple group comparison with post-hoc Bonferroni for all parameters with normal continuous data. For nonparametric data, the differences between the groups were assessed using the Kruskal–Wallis nonparametric test for multiple group comparison with the post-hoc Mann–Whitney test. *p* values are reported as Gaussian approximation. In order to evaluate possible factors associated with different chronic liver diseases (CHC and NASH), and to distinguish disease subgroups (CHC-NSF and CHC-SF), we performed logistic regression analysis. First univariate, then multivariate, modeling was applied with the Backward-Wald method. The odds ratio (OR), the 95% confidence interval of the odds ratio (95%CI OR), and the *p* value were reported. The diagnostic performance of plasma and tissue cytokines as biomarkers for distinguishing disease (CHC, NASH, CHC-NSF, CHC-SF) from healthy controls as well as between disease subgroups was performed, and the area under the ROC curve, *p* value, sensitivity, specificity, and recommended cut-off value were reported for all cytokines that showed the area under the ROC curve of at least 50%. Also, we presented the most valuable ROC curve for distinguishing CHC-NSF from CHC-SF patients. All statistical methods considered significant at the level of 0.05. A complete statistical analysis was performed in IBM SPSS ver. 26.

## 5. Conclusions

This study aimed to investigate the IL-23/IL-17 axis cytokine levels of both plasma and liver tissue in patients with different stages of CHC and NASH in order to elucidate the immunopathogenesis of these liver diseases. Anthropometric and biochemical characteristics of the patients are all in accordance with the etiology and the stage of liver disease. CHC patients with significant fibrosis and NASH patients have increased levels of IL-6, IL-17A and IL-23 in liver tissue as well as plasma, suggesting both local and systemic proinflammatory response. NASH patients have higher plasma levels of IL-17A and IL-23 cytokines compared to CHC with significant fibrosis and cirrhosis, as well as serum and tissue IL-17A compared to CHC without significant fibrosis but with same stages of fibrosis, reflecting probable higher systemic multifactorial inflammation in NASH compared to CHC. When it comes to tissue immune response, NASH and CHC patients with significant fibrosis and cirrhosis have higher local liver inflammation compared to systemic proinflammatory response.

Our research also establishes that a potential inflammatory biomarker for NASH could be high IL-23 plasma levels, while high levels of IL-17A and low levels of IL-10 could be used as plasma biomarkers of CHC. Importantly, the severity of fibrosis in CHC patients could be estimated based on the lower IL-10 levels in CHC-SF patients. We can conclude that proinflammatory pathways activated in CLDs such as CHC and NASH are important factors in the progression of disease, and understanding the role of specific cytokines and their potential as therapeutic targets could enhance therapeutic options, reduce the need for liver transplantation and decrease mortality.

## Figures and Tables

**Figure 1 ijms-24-12483-f001:**
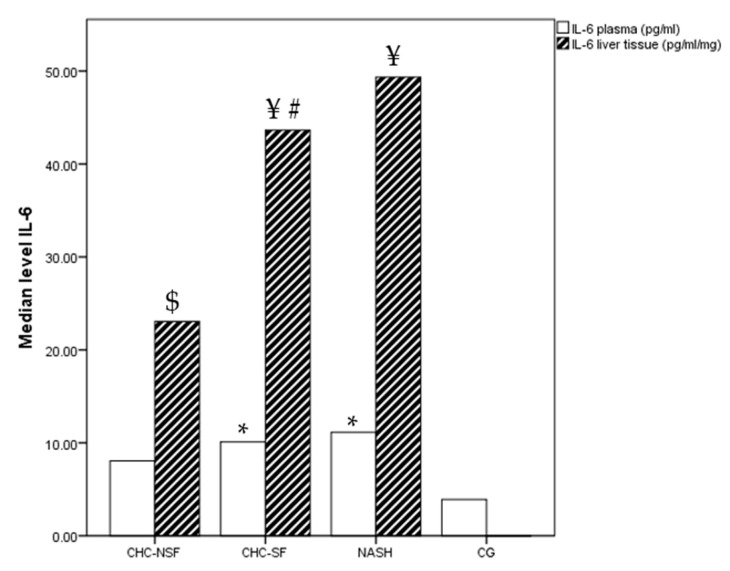
CHC-NSF: chronic hepatitis C with non-significant fibrosis, CHC-SF: chronic hepatitis C with significant fibrosis, NASH: non-alcoholic steatohepatitis, CG: control group, IL-6: interleukin 6. Median level of IL-6 in plasma and liver tissue of healthy controls and patients with liver diseases. Statistical difference was assessed using Kruskal–Wallis test. * *p* < 0.05 in comparison with CG; $ *p* < 0.05 in comparison with NASH; # *p* < 0.05 in comparison with CHC-NSF, ¥ *p* < 0.05 plasma levels in comparison with liver tissue levels (Post-hoc statistical difference using Mann–Whitney test); n (CG) = 20, n (NASH) = 19, n (CHC-NSF) = 20, n (CHC-SF) = 16.

**Figure 2 ijms-24-12483-f002:**
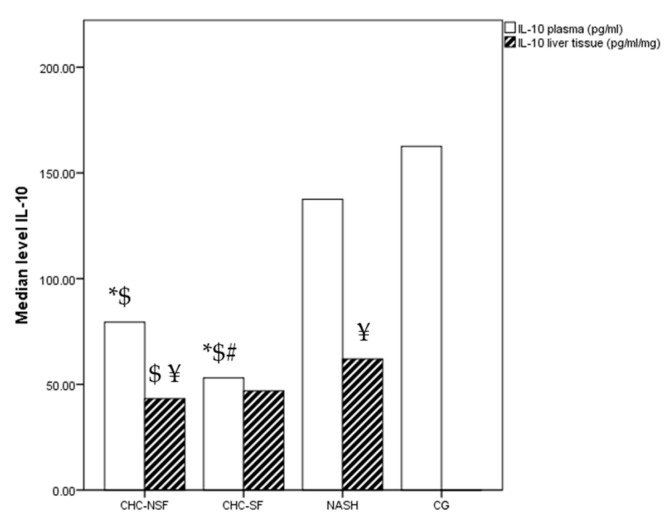
CHC-NSF: chronic hepatitis C with non-significant fibrosis, CHC-SF: chronic hepatitis C with significant fibrosis, NASH: non-alcoholic steatohepatitis, CG: control group, IL-10: interleukin 10. Median level of IL-10 in plasma and liver tissue of healthy controls and patients with liver diseases. Statistical difference was assessed using Kruskal–Wallis test. * *p* < 0.05 in comparison with CG; $ *p* < 0.05 in comparison with NASH; # *p* < 0.05 in comparison with CHC-NSF; ¥ *p* < 0.05 plasma levels in comparison with liver tissue levels (Post-hoc statistical difference using Mann–Whitney test); n (CG) = 20, n (NASH) = 19, n (CHC-NSF) = 20, n (CHC-SF) = 16.

**Figure 3 ijms-24-12483-f003:**
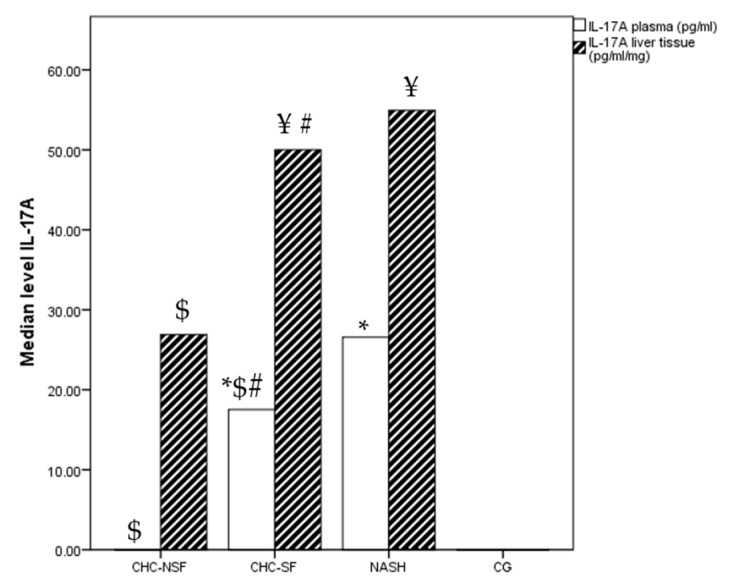
CHC-NSF: chronic hepatitis C with non-significant fibrosis, CHC-SF: chronic hepatitis C with significant fibrosis, NASH: non-alcoholic steatohepatitis, CG: control group, IL-17A: interleukin 17A. Median level of IL-17A in plasma and liver tissue of healthy controls and patients with liver diseases. Statistical difference was assessed using Kruskal–Wallis test. * *p* < 0.05 in comparison with CG; $ *p* < 0.05 in comparison with NASH; # *p* < 0.05 in comparison with CHC-NSF; ¥ *p* < 0.05 plasma levels in comparison with liver tissue levels (Post-hoc statistical difference using Mann–Whitney test); n (CG) = 20, n (NASH) = 19, n (CHC-NSF) = 20, n (CHC-SF) = 16.

**Figure 4 ijms-24-12483-f004:**
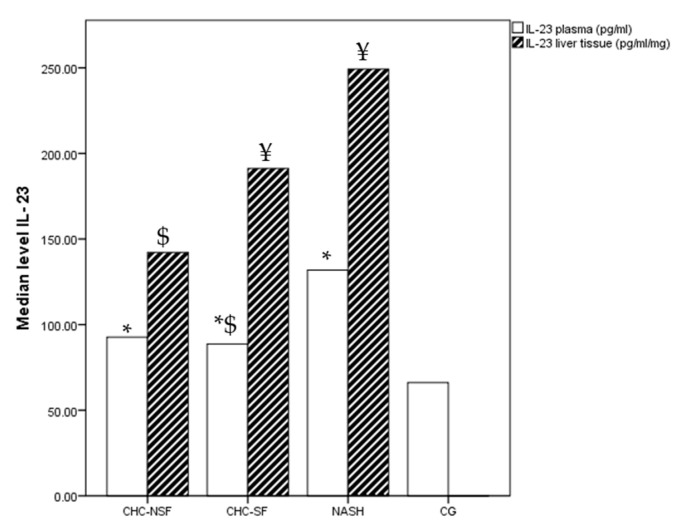
CHC-NSF: chronic hepatitis C with non-significant fibrosis, CHC-SF: chronic hepatitis C with significant fibrosis, NASH: non-alcoholic steatohepatitis, CG: control group, IL-23: interleukin 23. Median level of IL-23 in plasma and liver tissue of healthy controls and patients with liver diseases. Statistical difference was assessed using Kruskal–Wallis test. * *p* < 0.05 in comparison with CG; $ *p* < 0.05 in comparison with NASH; ¥ *p* < 0.05 plasma levels in comparison with liver tissue levels (Post-hoc statistical difference using Mann–Whitney test); n (CG) = 20, n (NASH) = 19, n (CHC-NSF) = 20, n (CHC-SF) = 16.

**Figure 5 ijms-24-12483-f005:**
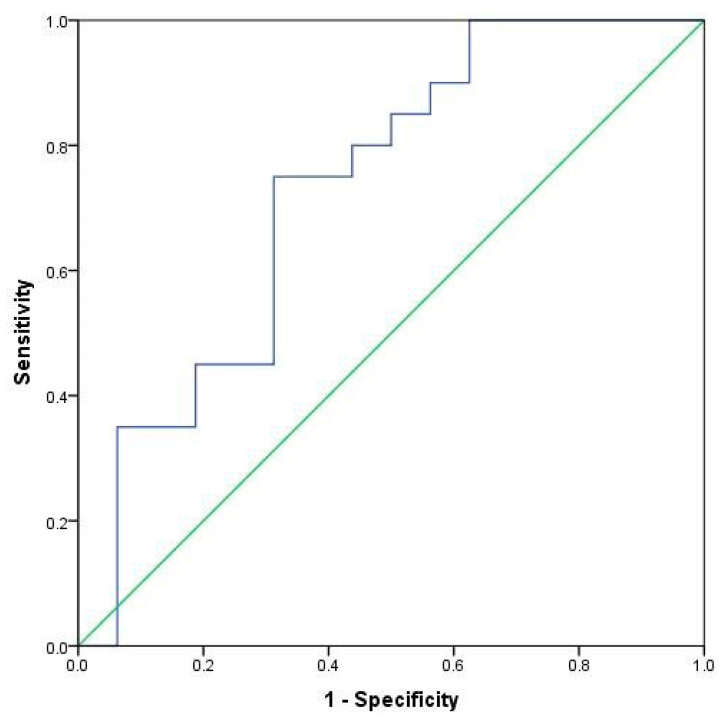
ROC curve of plasma IL-10 as biomarker for distinguishing CHC-NSF and CHC-SF patients. CHC-NSF: chronic hepatitis C with non-significant fibrosis, CHC-SF: chronic hepatitis C with significant fibrosis, IL: interleukin, blue line: ROC curve; green line: baseline.

**Table 1 ijms-24-12483-t001:** Demographic, anthropometric and biochemical characteristics in all groups of study subjects.

Variable	CHC-NSF (n = 20)	CHC-SF (n = 16)	NASH (n = 19)	CG (n = 20)	*p* Value
Age (years)	46.3 ± 13.6	54.2 ± 16.3	43.9 ± 9.2	42.5 ± 13.3	ns
Gender Female/Male (%)	49/51	52/48	34/66	47/53	ns
Body mass (kg)	70.0 ± 9.6	70.1 ± 9.5	96.7 ± 19.4 *#	67.8 ± 11.1	<0.001
Body height (cm)	177.8 ± 10.9	176.4 ± 11.6	179.0 ± 7.9	176.6 ± 11.6	ns
BMI (kg/cm^2^)	22.0 ± 0.9	22.4 ± 0.8	30.1 ± 5.5 *#	21.5 ± 1.2	<0.001
Erythrocytes (cells/mL)	4.9 ± 0.5	4.8 ± 0.6	5.0 ± 0.5	4.4 ± 0.5	ns
Hemoglobin (g/L)	147.7 ± 13.7	142.1 ± 19.3	144.3 ± 14.2	137.8 ± 7.6	ns
Leucocytes (cells/mL)	7.8 ± 2.5	6.5 ± 2.2	6.6 ± 1.6	5.6 ± 1.4	ns
PLT (cells/mL)	230.6 ± 63.2	182.2 ± 57.4 *#	220.1 ± 41.0	263.7 ± 66.7	<0.001
AST (U/L)	41.3 ± 39.9 *	104.5 ± 38.8 *#	50.1 ±15.7 *&	22.5 ± 5.6	<0.001
ALT (U/L)	61.1 ± 28.7 *	157.1 ± 29.9 *#	88.8 ± 44.2 *&	33.9 ± 11.1	<0.001
γGTP (U/L)	67.8 ± 94.0	81.0 ± 82.8	61.1 ± 45.9	24.3 ± 5.1	ns
AP (u/L)	66.6 ± 24.9	80.0 ± 53.9	67.1 ± 17.5	71.4 ± 14.3	ns
AFP (ng/mL)	4.4 ± 2.4	11.8 ± 8.2 *#	2.9 ± 1.1 &	1.9 ± 0.6	<0.001
TSH (microU/mL)	1.5 ± 0.8	1.6 ± 0.6	1.7 ± 0.9	2.5 ± 0.5	ns
Glucose (mmol/L)	5.7 ± 1.5	5.8 ± 1.1	5.4 ± 0.7	4.7 ± 0.5	ns
Uric acid (µmol/L)	5.3 ± 2.1	6.2 ± 1.6	5.6 ± 1.5	4.7 ± 1.2	ns
Creatinine (µmol/L)	79.8 ± 19.6	76.6 ± 17.2	81.5 ± 14.6	67.4 ± 10.8	ns
Total cholesterol (mmol/L)	4.7 ± 1.1	4.2 ± 1.0	7.1 ± 0.8 *#&	3.6 ± 0.7	<0.001
LDL (mmol/L)	2.6 ± 1.0	2.4 ± 0.8	4.7 ± 0.8 *#&	2.5 ± 0.5	<0.001
HDL (mmol/L)	1.5 ± 0.5 *	1.4 ± 0.3 *	1.3 ± 0.7 *	2.0 ± 0.5	<0.001
Triglycerides (mmol/L)	1.25 ± 0.8	1.20 ± 0.8	4.13 ± 1.13 *#&	1.14 ± 0.2	<0.001
Total proteins (g/L)	76.6 ± 6.5	75.1 ± 7.6	76.3 ± 7.7	75.9 ± 6.1	ns
Albumin (g/L)	38.9 ± 3.1 *	31.1 ± 2.4 *#	43.3 ± 5.5 &	42.6 ± 3.9	<0.001
Total bilirubin (mg/dL)	10.6 ± 4.5	10.9 ± 2.7	13.7 ± 9.3	10.5 ± 3.3	ns
Direct bilirubin (mg/dL)	3.9 ± 1.8	4.6 ± 1.4	3.9 ± 1.5	2.1 ± 1.5	ns
INR	1.0 ± 0.2	1.1 ± 0.2	1.0 ± 0.1	1.0 ± 0.1	ns
Fibrinogen (mg/dL)	2.9 ± 0.5	2.8 ± 0.5	3.1 ± 0.4	2.6 ± 0.4	ns
CRP (mg/dL)	2.5 ± 2.0	3.7 ± 2.0	3.8 ± 2.3	1.3 ± 0.6	ns

Data are given as mean ± SD; CHC-NSF: chronic hepatitis C with non-significant fibrosis, CHC-SF: chronic hepatitis C with significant fibrosis, NASH: non-alcoholic steatohepatitis, CG: control group, BMI: body mass index, PLT: platelets, AST: aspartate aminotransferase, ALT: alanine aminotransferase, γGTP: γ-glutamyl transpeptidase; AP: alkaline phosphatase; AFP: alpha fetoprotein; TSH: thyroid-stimulating hormone; LDH: lactate dehydrogenase; CK: creatine kinase; LDL: low-density lipoprotein cholesterol; HDL: high-density lipoprotein cholesterol; INR: indicates international normalized ratio; CRP: c-reactive protein; ns: not significant. Statistical difference was assessed using ANOVA test. * *p* < 0.05 in comparison with CG; # *p* < 0.05 in comparison with CHC-NSF; & *p* < 0.05 in comparison with CHC-SF (Post-hoc statistical difference using Bonferroni test).

**Table 2 ijms-24-12483-t002:** Logistic regression analysis in all study groups.

Variable	OR	95%CI OR	*p* Value	Adjusted *p* Value
CG vs. CHC				
Plasma IL-10	0.974	0.96–0.99	0.001	0.001
Plasma IL-17A	1.176	1.03–1.34	0.016	0.035
CG vs. NASH				
Plasma IL-23	1.026	1.01–1.04	0.005	0.009
CG vs. CHC-NSF				
Plasma IL-10	0.984	0.97–0.99	0.012	0.014
CG vs. CHC-SF				
Plasma IL-17A	1.566	1.14–2.14	0.005	0.005
NASH vs. CHC				
Plasma IL-17A	0.915	0.86–0.98	0.007	0.005
NASH vs. CHC-NSF				
Plasma IL-17A	0.900	0.84–0.97	0.004	0.007
NASH vs. CHC-SF				
Plasma IL-17A	0.763	0.61–0.96	0.021	0.020
CHC-NSF vs. CHC-SF				
Plasma IL-10	0.983	0.97–0.99	0.013	0.032

CHC: chronic hepatitis C, CHC-NSF: chronic hepatitis C with non-significant fibrosis, CHC-SF: chronic hepatitis C with significant fibrosis, NASH: non-alcoholic steatohepatitis, CG: control group, IL: interleukin, OR: odds ratio.

**Table 3 ijms-24-12483-t003:** Diagnostic performances of cytokines in differentiation of study subgroups.

Variable	Area under the Curve (%), *p* Value	Recommended Cut-Off	Sn (%)	Sp (%)
CG vs. CHC				
Plasma IL-6	67.8, 0.028	≥4.12	72.2	55.0
Plasma IL-17A	71.7, *p* = 0.008	≥2.68	52.8	85.0
Plasma IL-23	69.0, *p* = 0.019	≥47.32	91.7	40.0
CG vs. CHC-NSF				
Plasma IL-23	70.0, *p* = 0.030	≥66.14	75.0	50.0
CG vs. CHC-SF				
Plasma IL-6	74.4, 0.013	≥6.09	81.3	65.0
Plasma IL-17A	99.1, *p* < 0.001	≥3.32	100.0	85.0
CHC-NSF vs. CHC-SF				
Plasma IL-10	72.8, 0.020	≥47.00	90.0	43.7

CHC: chronic hepatitis C, CHC-NSF: chronic hepatitis C with non-significant fibrosis, CHC-SF: chronic hepatitis C with significant fibrosis, CG: control group, IL: interleukin, Sn: sensitivity, Sp: specificity.

## Data Availability

All data will be available in Database of Faculty of Medicine, University of Belgrade (http://med.bg.ac.rs/).

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
