# Peer review of "IL-23/IL-17 Axis in Chronic Hepatitis C and Non-Alcoholic Steatohepatitis—New Insight into Immunohepatotoxicity of Different Chronic Liver Diseases"

_ijms, 2023, doi:10.3390/ijms241512483_

Round 1

Reviewer 1 Report

It is an interesting manuscript about “ L-23/IL-17 axis in the Chronic Hepatitis C and Nonalcoholic Steatohepatitis – New Insight Into Immuno-hepatotoxicity of Different Chronic Liver Diseases ” 

My concern is determined in the following points.

Liver tissue levels of IL-17A and IL-23 in CHC-NSF were significantly lower in compared with NASH. IN CHC-SF and NASH, IL17-A and IL-23 in liver tissue were significantly higher in compare to plasma levels. 

Proinflammatory response of IL-23/17A axis is dominant in plasma and liver tissue in CHC with higher levels of liver fibrosis and in NASH 

patients. Proinflammation in CLD such as CHC and NASH, is one of important factors that lead in progression of disease and can be potential therapeutic option which will reduce the need for liver transplantation and mortality. 

Change over time of plasma ( and/or liver tissue) levels of IL17A/IL23 in patients with chronic hepatitis C during and after treatment should be revealed.

Interleukin-23 (IL-23) and its downstream factor IL-17 are the key cytokines involved in immune and inflammatory response in chronic liver diseases. The IL-23/Th17 axis plays an important role in development of chronic HCV infection and antiviral response. IL-23 may enhance the antiviral activity of interferon-based therapy by modulating the expression of Th17 cells-associated molecules in HCV-infected patients.

The immunopathologic and prognostic role of the Th17/IL-17 axis and related pathways in fibrogenesis and progression to cirrhosis in patients with liver disease due to HCV.

Above mentioned factors should be referred to.

Author Response

Dear Reviwer 1,

Thank You very much for giving us the opportunity to resubmit our manuscript. We would like to thank You for your valuable time, feedback and comments with regards to our submission.

Reviewer 1: Liver tissue levels of IL-17A and IL-23 in CHC-NSF were significantly lower in compared with NASH. IN CHC-SF and NASH, IL17-A and IL-23 in liver tissue were significantly higher in compare to plasma levels. Proinflammatory response of IL-23/17A axis is dominant in plasma and liver tissue in CHC with higher levels of liver fibrosis and in NASH patients. Proinflammation in CLD such as CHC and NASH, is one of important factors that lead in progression of disease and can be potential therapeutic option which will reduce the need for liver transplantation and mortality. Change over time of plasma (and/or liver tissue) levels of IL17A/IL23 in patients with chronic hepatitis C during and after treatment should be revealed.

Author: Dear reviewer, thank You very much for the support and for more than useful comments that helps us a lot with manuscript improvement. In this study we couldn’t follow immunological responses during and after therapy because of in our country in that moment DAA therapy wasn’t available and all of our CHC-NSF patients were not motivated for peg-IFN therapy.

Reviewer 1: Interleukin-23 (IL-23) and its downstream factor IL-17 are the key cytokines involved in immune and inflammatory response in chronic liver diseases. The IL-23/Th17 axis plays an important role in development of chronic HCV infection and antiviral response. IL-23 may enhance the antiviral activity of interferon-based therapy by modulating the expression of Th17 cells-associated molecules in HCV-infected patients.

Author: Thank You very much for this suggestion, we added an importance of immunological response on antiviral therapy (line 326-330).

Reviwer 1: The immunopathologic and prognostic role of the Th17/IL-17 axis and related pathways in fibrogenesis and progression to cirrhosis in patients with liver disease due to HCV.

Author: Thank You once again for all suggestions and comments. We have detailed described role of the Th17/IL-17 axis in discussion part (line 309-321).

Once again, thank you for all suggestions and comments that helped us a lot to improve the quality of the paper.

Reviewer 2 Report

Authors Vujovic et al. have submitted a manuscript detailing a study in patients with chronic liver diseases of 2 etiologies, viral hepatitis and NASH, compared to controls to describe the levels of cytokines in plasma and tissue.  Unfortunately the study as presented is too descriptive and lacking in much more insight for the respective diseases.  Although studies looking into the immunophenotype of various liver diseases have merit, such that understanding which factors might play pivotal roles in disease progression, the scope of this study is extremely limited.

NASH and NAFLD patients exist on a spectrum and the inflammatory components of the disease are far more complex than this characterization allows.  Moreover, many of these factors have likely been described in great detail previously, in patient populations and in various preclinical models with knockout mice.  Greater insight into how these cytokines relate to biochemical markers measured in Table 1 would help add sophistication to these studies.  This is similar to CHC patients, with or without fibrosis.  Distinguishing the differences between these populations can be of use to clinicians and therapeutic identification, but not as presented in its current form.  

Overall, I think the authors need to perform multiple types of regressions to see how these plasma or tissue levels could distinguish anything about the patient populations as well as add some preclinical data to support these findings mechanistically.  

The manuscript is understandable, but several typos and formatting errors (hyphenated words) can be found.  Moreover, the introduction is just one long paragraph that likely needs to be broken up into more discrete paragraphs detailing the reasons for the study at greater length.  

Author Response

Dear Reviwer 2,

Thank You very much for giving us the opportunity to resubmit our manuscript. We would like to thank You for your valuable time, feedback and comments with regards to our submission.

Reviewer 2: NASH and NAFLD patients exist on a spectrum and the inflammatory components of the disease are far more complex than this characterization allows.  Moreover, many of these factors have likely been described in great detail previously, in patient populations and in various preclinical models with knockout mice.

Author: Dear reviewer, thank You very much for the support and for more than useful comments that helps us a lot with manuscript improvement. We respected them all and edited the manuscript in accordance with Your suggestions. We agree with You that experimental models are very important for understanding imunopathogenesis of these diseases, so we point this studies in introduction, line 97-107, and in discussion part line 323-326 and 412-416.

Reviewer 2: Greater insight into how these cytokines relate to biochemical markers measured in Table 1 would help add sophistication to these studies.  This is similar to CHC patients, with or without fibrosis.  Distinguishing the differences between these populations can be of use to clinicians and therapeutic identification, but not as presented in its current form. 

Author: Thank You very much for this suggestion, we agree that from previous form of manuscript couldn’t enhance relationship between cytokines and biochemical markers, so we added statistical analyzes with Spirman’s correlation test, which is presented in Result part (line 229-238) and discussed in Discussion part (line 424-443).

Reviewer 2: Overall, I think the authors need to perform multiple types of regressions to see how these plasma or tissue levels could distinguish anything about the patient populations as well as add some preclinical data to support these findings mechanistically. 

Author: Thank You very much for this suggestion. In order to distinguish main study groups (CHC and NASH) from healthy population (CG), as well as, to determine the difference between CHC subgroups (NSF and SF) according to plasma/tissue cytokine concentrations we performed logistic regression analysis and presented results in Table 2 (line 240-259).

Reviewer 2: The manuscript is understandable, but several typos and formatting errors (hyphenated words) can be found.  Moreover, the introduction is just one long paragraph that likely needs to be broken up into more discrete paragraphs detailing the reasons for the study at greater length

Author: Thank You once again for all suggestions and comments that helped us a lot to improve the quality of the paper. We completed introduction with explanation why this axis was challenging for us- introduction part line 79-109.

In addition, the paper was re-checked for language errors by native English speaker and all technical mistakes are corrected throughout the article text.

Once again, thank you for all suggestions and comments that helped us a lot to improve the quality of the paper.

Round 2

Reviewer 2 Report

I thank the authors for their revised manuscript concerning the plasma and liver levels of the IL-23/IL-17 axis in chronic liver diseases.  This study has the potential to look at how soluble factors might differentiate etiologies of chronic liver disease, with a focus on whether significant fibrosis is present.  By extension, this could support noninvasive biomarker identification or a diagnostic tool for patients instead of biopsy.  

The authors did an extensive rework of the introduction and included some new regression analysis in Table 2, both of which a welcomed additions. 

The revision from first iteration of this study is a significant improvement, but there is more that can be done to present this data and study to merit acceptance.  

Minor: The data in Table 2 require a legend to indicate what is presented.  

Major: The binary comparisons of each liver disease are a good start to demonstrating the utility of this study.  More complex analyses with the data are still required.  As in: How do these cytokine levels within the groups correlate to the data from Table 1?  Is there predictive power in the clustered data of all cytokines and each disease?  ROC analysis should be attempted and curves in another figure, for example. 

These data, albeit a likely first time presented together, or from a comprehensively detailed cohort, are new but are probably supported from various preclinical studies. More analysis is necessary to support why this study is important.

Several instances of subject-verb agreement, tenses and awkward sentence structures are found throughout.  Most of these are found in the reworked abstract and introduction.  Please take special care in revising written English. 

Author Response

Dear Editor,

Thank You very much for giving us the opportunity to resubmit our manuscript entitled “IL-23/IL-17 axis in the Chronic Hepatitis C and Nonalcoholic Steatohepatitis – New Insight Into Immunohepatotoxicity of Different Chronic Liver Diseases” for the publication in the International Journal of Molecular Science – Special Issue “Molecular Mechanisms of Hepatotoxicity 2.0”. Thank You for the more than useful reviewers’ comments. We would like to thank You and the reviewers for your valuable time, feedback and comments with regards to our submission. We read carefully all reviewers comments and used this opportunity to submit response. The paper was re-checked for language errors by native English speaker.

Reviewer 2 comments (Round 2):

Reviewer 2: The data in Table 2 require a legend to indicate what is presented.

Author: Dear reviewer, thank You very much for this minor suggestion, we added legend for all tables and figures.

Reviewer 2: The binary comparisons of each liver disease are a good start to demonstrating the utility of this study.  More complex analyses with the data are still required.  As in: How do these cytokine levels within the groups correlate to the data from Table 1?  Is there predictive power in the clustered data of all cytokines and each disease?  ROC analysis should be attempted and curves in another figure, for example. These data, albeit a likely first time presented together, or from a comprehensively detailed cohort, are new but are probably supported from various preclinical studies. More analysis is necessary to support why this study is important.

Author: Thank You very much for this suggestion, we added correlation analyzes in Result part (line 273-299), as well as diagnostic accuracy analysis with ROC curve (line 324-346) and discussed all results.

In addition, the paper was once again re-checked for language errors by native English speaker and all technical mistakes are corrected throughout the article text.

Once again, thank you for all suggestions and comments that helped us a lot to improve the quality of the paper.

Round 3

Reviewer 2 Report

I thank the authors for another quick turnaround of this manuscript to include both the regression and ROC analysis of their study.  The extensive editing is noted and overall presents as a more complete study of the IL-23/IL-17 axis in various liver diseases.  The authors should be careful in overstating conclusions, however, like line 490-491 "All of these results are in correlation with the results of our study, so we can point that IL-17A contribute to chronic hepatitis-induced liver fibrosis."  This statement is quite definitive, yet the entirety of this study is associative.  The authors cannot conclude that IL-17A contributes to fibrosis without significant preclinical studies and patient cohorts along a spectrum of disease.  Please edit these to demonstrate that a strong association has been noted, but not a causative link.  

I have no further comments.  

There are a few disjointed sentences and grammatical errors that persist, possibly due to the extensive editing to this manuscript.  I think this can be fixed during proofing, but authors should make every effort to edit the text for grammar.  

Author Response

Dear Editor,

Thank You very much for giving us the opportunity to resubmit our manuscript entitled “IL-23/IL-17 axis in the Chronic Hepatitis C and Nonalcoholic Steatohepatitis – New Insight Into Immunohepatotoxicity of Different Chronic Liver Diseases” for the publication in the International Journal of Molecular Science – Special Issue “Molecular Mechanisms of Hepatotoxicity 2.0”. The paper was re-checked for language errors by native English speaker.

Reviewer 2 comments (Round 3):

Reviewer 2: The authors should be careful in overstating conclusions, however, like line 490-491 "All of these results are in correlation with the results of our study, so we can point that IL-17A contribute to chronic hepatitis-induced liver fibrosis."  This statement is quite definitive, yet the entirety of this study is associative.  The authors cannot conclude that IL-17A contributes to fibrosis without significant preclinical studies and patient cohorts along a spectrum of disease.  Please edit these to demonstrate that a strong association has been noted, but not a causative link.   

Author: Dear reviewer, thank You very much for this suggestion, we agree with You that we cannot conclude this association without more studies, so we changed this sentence in line 491-495.

In addition, the paper was once again re-checked for language errors by native English speaker and all technical mistakes are corrected throughout the article text. Finally it will be re-checked during proof version.

Once again, thank You for all suggestions and comments that helped us a lot to improve the quality of the paper.
